# Characterization of the virome of shallots affected by the shallot mild yellow stripe disease in France

**Armelle Marais**⦿*, **Chantal Faure, Sébastien Theil, Thierry Candresse**

UMR 1332, Biologie du Fruit et Pathologie, INRA, Université de Bordeaux, Villenave d'Ornon, France

* armelle.marais-colombel@inra.fr

**Data Availability Statement:** All relevant data are within the manuscript and its Supporting Information files.

**Funding:** The authors received no specific funding for this work.

## Abstract

To elucidate the etiology of a new disease of shallot in France, double-stranded RNAs from asymptomatic and symptomatic shallot plants were analyzed using high-throughput sequencing (HTS). Annotation of contigs, molecular characterization and phylogenetic analyses revealed the presence in symptomatic plants of a virus complex consisting of shallot virus X (ShVX, *Allexivirus*), shallot latent virus (SLV, *Carlavirus*) and two novel viruses belonging to the genera *Carlavirus* and *Potyvirus*, for which the names of shallot virus S (ShVS) and shallot mild yellow stripe associated virus (SMYSaV), are proposed. Complete or near complete genomic sequences were obtained for all these agents, revealing divergent isolates of ShVX and SLV. Trials to fulfill Koch's postulates were pursued but failed to reproduce the symptoms on inoculated shallots, even though the plants were proved to be infected by the four viruses detected by HTS. Replanting of bulbs from SMYSaV-inoculated shallot plants resulted in infected plants, showing that the virus can perpetuate the infection over seasons. A survey analyzing 351 shallot samples over a four years period strongly suggests an association of SMYSaV with the disease symptoms. An analysis of SMYSaV diversity indicates the existence of two clusters of isolates, one of which is largely predominant in the field over years.

## Introduction

The economically important cultivated *Allium* species are garlic (*Allium sativum*), leek (*Allium ampeloprasum* var. *porrum*), onion (*Allium cepa*), and its relative shallot (*Allium cepa* L. var. *aggregatum*) [1]. Shallot is mainly cultivated for culinary purposes, while onion and garlic are also used in traditional medicine. Viral infections are a significant problem for all *Allium* crops, even more so in the case of shallot and garlic which are exclusively vegetative propagated, leading to the accumulation of viruses in planting material [2]. Due to their prevalence and the damages they cause, the most economically important *Allium* viruses are members of the genus *Potyvirus*, particularly onion yellow dwarf virus (OYDV) and leek yellow stripe virus (LYSV). In the Mediterranean basin, shallot yellow stripe virus (SYSV) and turnip mosaic virus (TuMV) have also been described infecting *Allium* species, as well as four other

**Competing interests:** The authors have declared that no competing interests exist.

potyviruses of lower incidence, even though TuMV has not been reported on shallot so far [2]. Potyviruses and carlaviruses are frequently found on cultivated *Allium* crops and are transmitted non persistently by aphids. The first described *Allium*-infecting carlavirus was shallot latent virus (SLV, synonym with garlic latent virus, GLV), which seems to be asymptomatic in garlic, onion and shallot when in single infection but can cause significant yield losses in the presence of potyviruses due to synergistic effects [3]. Another carlavirus (garlic common latent virus, GarCLV) is frequently detected on garlic, onion and leek, associated with symptomless infection. Eight viral species belonging to the genus *Allexivirus* in the family *Alphaflexiviridae* have also been described from *Allium* species. Only two of them, shallot virus X (ShVX) and shallot mite-borne latent virus (SMbLV) have been described in shallot, in which they cause latent infections. All allexiviruses are transmitted by mites and coinfections with potyviruses and carlaviruses are frequent, with potential synergistic effects that could lead to increased damages [2]. Besides the viruses belonging to the *Allexivirus*, *Potyvirus*, and *Carlavirus* genera, five other viruses infecting *Allium* species have been described, generally with limited incidence, including iris yellow spot virus (IYSV), a member of the genus *Orthotospovirus*, reported on shallot [4].

In 2012, a new disease was observed in the west of France in shallots. Symptoms consisted of yellow stripes on the leaves, associated with a loss of vigor, considered as moderate as compared to that caused by OYDV or LYSV (Fig 1). This gave its name to the disease, shallot mild yellow stripe disease (SMYSD). Early tests revealed that the new disease could be observed in plants that test negative for OYDV and LYSV, indicating that these two viruses were not involved. In parallel, meristem-tip culture from symptomatic plants led to the disappearance of the symptoms, reinforcing the hypothesis of a viral etiology. The present study was therefore initiated with the objective to identify the causal agent(s) of this new disease.

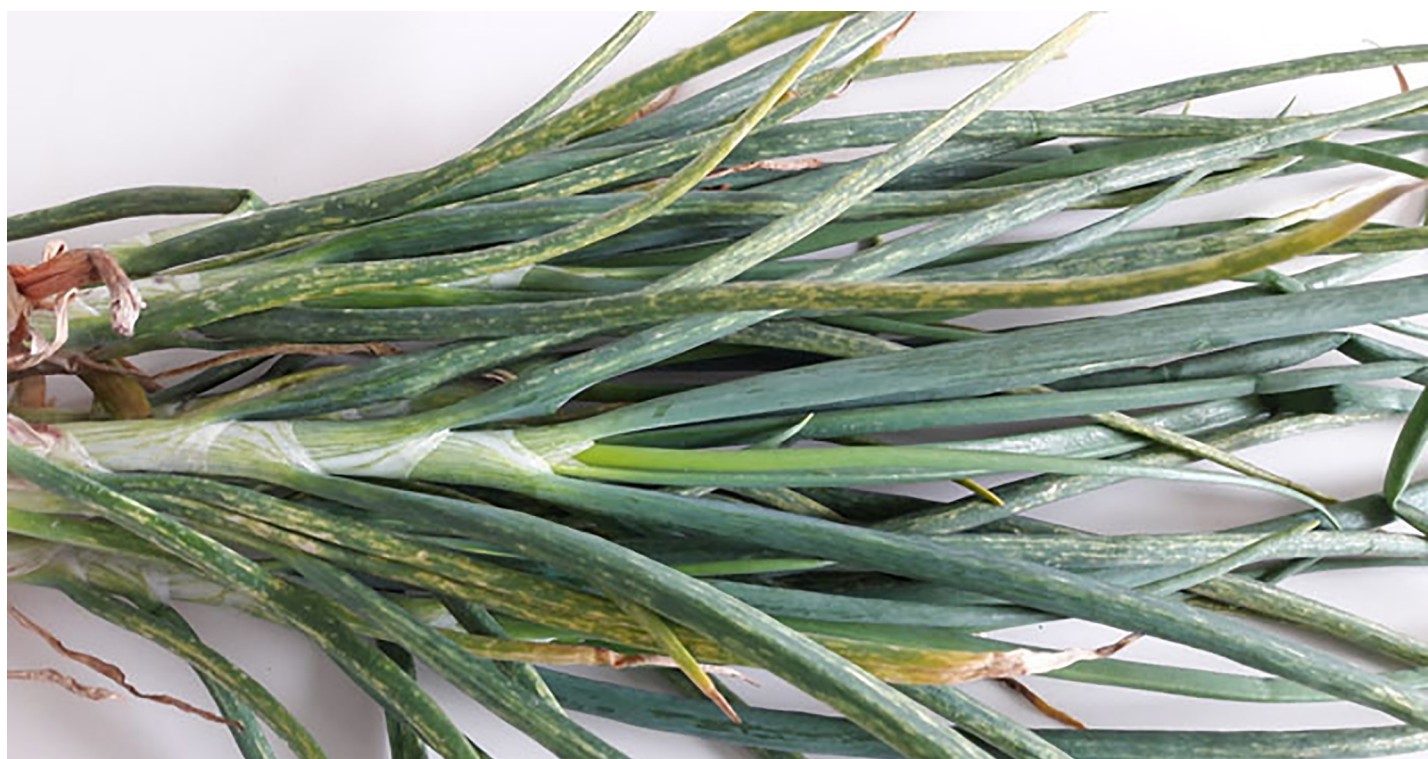

**Fig 1. Symptoms associated with the shallot mild yellow stripe disease on shallot plant.**

## Materials and methods

### Plant samples

Six samples (13–01 to 13–06) of shallot (*Allium cepa* L. var. *aggregatum*) were collected in West France in 2013 and analyzed using high-throughput sequencing (HTS). Two of these samples were from asymptomatic plants while the other four showed symptoms of the novel SMYS disease. The two symptomatic samples 13–04 and 13–05 were collected in the same field, while the four remaining samples (one symptomatic and the two asymptomatic ones) were from distinct locations. In addition to these samples analyzed by HTS, a total of 351 symptomatic or asymptomatic shallot samples was collected over a four-year period (2014 to 2017) and screened for the presence of OYDV, LYSV and for the novel viruses detected in the present study. The symptoms observed on these plants were recorded using a 0 to 3 notation scale for both leaf striping and loss of vigor. The "0" score is defined as no symptom, the "3" score is defined as a symptomatology equivalent to that observed on control plants infected by OYDV. The "1" and "2" scores are used for symptoms of intermediate intensity. The study was carried out on private lands which the owner gave permission to conduct the study.

### Illumina sequencing of double-stranded RNAs from shallot samples

Double-stranded RNAs (dsRNAs) were purified from the two asymptomatic plant samples (13–01 and 13–02) and the four symptomatic ones (13–03 to 13–06), following the protocol previously described [5]. After reverse transcription and random amplification, the obtained cDNAs were used for the preparation of libraries and sequenced in multiplex format (Illumina HiSeq 2000 in paired end 2x 100 nt reads). The raw data are available on portail data INRA at https://doi.org/10.15454/6XCMAE. After quality trimming and demultiplexing steps [6], reads were assembled into contigs which were annotated by BlastN and BlastX comparisons [7] against the GenBank database using CLC Genomics Workbench 8. When needed, contigs corresponding to particular agents were further extended by several rounds of mapping of unassembled reads and/or assembled manually into scaffolds by alignment against reference viral genomes identified during the Blast analyses.

### Total nucleic acids extraction and detection of selected viruses by RT-PCR

Total nucleic acids were extracted from shallot samples and from the test plants of the host range experiments using the silica-capture procedure 2 described by [8]. The viruses were detected by two-step RT-PCR assays, following the procedure already described [9] and using specific primers (S1 Table). The amplified fragments were visualized on non-denaturing 1% agarose gels and, if needed, submitted to direct Sanger sequencing on both strands (GATC Biotech, Mulhouse, France).

### Completion of the genome sequences of the novel viruses and of divergent isolates of shallot latent virus (SLV) and shallot virus X (ShVX)

The 5' ends of the viral genomes sequences were determined or confirmed using the 5' Rapid Amplification of cDNA Ends (RACE) strategy and internal primers designed from the genomic contigs (S1 Table) following the kit supplier's recommendations (Takara Bio Europe/ Clontech, Saint Germain-en-Laye, France). The 3' ends were amplified using forward internal and polyA-anchored LD primers (S1 Table) as described [10]. Internal gaps and regions of low coverage were determined or confirmed by direct sequencing of RT-PCR fragments obtained using internal primers designed from the contigs (S1 Table). All amplified fragments were

visualized on non-denaturing agarose gels and directly sequenced on both strands by Sanger sequencing (GATC Biotech).

## Sequence analysis, comparisons, and phylogenetic analyses

Phylogenetic and molecular evolutionary analyses were conducted using MEGA version 6 [11]. Multiple alignments of nucleotide or amino acid sequences were performed using the ClustalW program [12] as implemented in MEGA6. Phylogenetic trees were reconstructed using the neighbor-joining method with strict nucleotide or amino acid distances and randomized bootstrapping for the evaluation of branching validity. Mean diversities and genetic distances (p-distances calculated on nucleotide or amino acid identity) were calculated using MEGA6.

## Mechanical inoculation of both novel viruses to indicator plants

A mixture of leaves from four plants identified as infected by the novel potyvirus but not by the novel carlavirus, OYDV or LYSV was used as the first inoculum. Similarly, a mixture of leaves from four plants known to be infected by the new carlavirus but free of the new potyvirus, of OYDV or LYSV was used as the second inoculum. All pools of leaves were ground 1:4 (wt/vol) in a solution of 0.03 M $Na_2HPO_4$ containing 0.2% sodium diethyldithiocarbamate (DIECA), and 100 mg each of carborundum and activated charcoal were added before rub-inoculation. A total number of 59 *Nicotiana benthamiana* plants, 48 *Chenopodium quinoa*, 34 *C. amaranticolor*, 31 *N. occidentalis*, and 23 *N. tabacum* cv xanthi were evaluated as potential hosts for the novel potyvirus and 23 plants of each species were used for trials involving the new carlavirus. The appearance of symptoms was monitored over a three-week period. At the end of the experimentation, the presence of the virus(es) in non-inoculated parts of the test plants was assessed by specific RT-PCR assays.

## Trials to fulfill Koch's postulates

Koch's postulate was evaluated separately for the two novel viruses and for a complex of four viruses (ShVX, SLV and the two novel viruses). A total of 21 virus-free shallots grown from seeds were mechanically inoculated with a mix of four plants shown to be co-infected by ShVX, SLV and the novel carlavirus and potyvirus, as described above. Plants were monitored for symptoms appearance over a five weeks period post inoculation. At the end of this period, the plants were tested by specific RT-PCR for the presence of the four inoculated viruses. In parallel, inoculation of shallot and onion plants was performed with an inoculum constituted of a pool of four plants known to be infected by the sole new potyvirus (63 and 40 plants of each *Allium* species, all grown from seeds) or infected only by the novel carlavirus (36 and 23 plants, respectively). A mix of leaves from two plants infected with the sole OYDV was used as a positive control for mechanical inoculation of shallot and onion plants. Bulbs from all inoculated shallot plants were replanted in insect-proof greenhouse and the resulting plants observed over an eight months period and tested for the presence of inoculated viruses.

## Results

### Illumina sequencing of double-stranded RNAs from asymptomatic and symptomatic shallot samples

After demultiplexing, quality trimming, and *de novo* assembly, BlastN and BlastX comparisons of the contigs obtained with the GenBank database showed that all sources but one (13–01) were infected by more than one viral species (Table 1). For the 13–01 asymptomatic sample,

**Table 1. Number and percentages of high-throughput sequencing reads (73 nucleotides average length) of shallot virus X (variants 1 and 2), shallot latent virus, the novel carlavirus and the novel potyvirus in each sample analyzed by Illumina sequencing.**

| Sample [a] | Total reads [b] | Shallot virus X | | Shallot latent virus | Novel carlavirus | Novel potyvirus |
|---|---|---|---|---|---|---|
| | | Variant 1 | Variant 2 | | | |
| 13–01 AS | 328,460 | 151,580 (46%) MH389253 [c] | 22,801 (6.9%) MH389254 [e] | | | |
| 13–02 AS | 1,412,128 | | | 249,097 (17.6%) MH389247 | 27,786 (1.9%) | |
| 13–03 S | 505,315 | | | 236,815 (46.9%) MH389249 [f] | 13,684 (2.7%) | 105,209 (20.8%) |
| 13–04 S | 438,574 | 60,019 (13.7%) MH389255 [d] | 158,755 (36.2%) MH389250 | | 86,627 (19.7%) | 48,064 (11%) |
| 13–05 S | 360,248 | 139,070 (38.6%) MH389251 | | | 81,900 (22.7%) MH292861 | 43,683 (12.1%) |
| 13–06 S | 778,696 | 34,273 (4.4%) MH389252 | | 257,193 (33%) | | 86,044 (11%) MG910502 |

Relevant GenBank accession numbers are indicated

[a] AS asymptomatic; S symptomatic

[b] After quality trimming

[c] genome sequence lacks 53 nt at the 5' end and 443 nt at the 3' end

[d] genome sequence lacks 7 nt at the 5' end and 111 nt at the 3' end

[e] genome sequence lacks 53 nt at the 5' end and 330 nt at the 3' end

[f] genome sequence lacks 84 nt at 5' end and 34 nt at 3' end

174,381 reads were assembled into contigs with high homology to isolates of shallot virus X (ShVX, genus *Allexivirus*, family *Alphaflexiviridae*). Two variants of ShVX were identified and reassembled from that plant, differing by their level of nucleotide (nt) identity with known ShVX sequences. Most of the reads (151,580) were assembled into contigs (hereafter referred to as ShVX 13–01 variant 1) closely related to ShVX isolate JX310755 (97–98% of nt identity depending on the contigs). More divergent contigs (hereafter referred to as ShVX 13–01 variant 2) integrating 22,801 reads could also be assembled. They showed between 82 and 90% of nt identity with JX310755, depending on the contig (Table 1). In the other asymptomatic sample (13–02), two viruses were detected: a divergent isolate of shallot latent virus (SLV, genus *Carlavirus*, family *Betaflexiviridae*), integrating 249,097 reads and sharing around 83% of nt identity with reference SLV isolates, and a putative novel carlavirus. Indeed, a total of 27,786 reads (corresponding to 1.9% of the total reads) were assembled into contigs sharing relatively weak nt identities (71–75% depending on the contig) with various carlaviruses. In the four symptomatic samples (13–03 to 13–06), besides the presence of one or more of the above viruses (Table 1), contigs integrating between 43,683 and 105,209 reads depending on the sample and showing at most 74% of nt identity with leek yellow stripe virus (LYSV, genus *Potyvirus*, family *Potyviridae*) were detected, leading to the hypothesis of the presence of a novel potyvirus.

In the end, the complete genomic sequences of seven viral isolates were obtained (Table 1): ShVX variant 1 from samples 13–05 and 13–06, ShVX variant 2 from sample 13–04, SLV from samples 13–02 and 13–06, the novel carlavirus from sample 13–05 and the novel potyvirus from sample 13–06. Moreover, near complete genome sequences of three additional ShVX isolates (two from sample 13–01, and one from sample 13–04) and of one additional SLV isolate (from sample 13–03) were also obtained during the assembly process (Table 1) but no specific effort was made to complete their missing 5' and 3' genome ends.

Besides the whole genome sequence determined for the novel carlavirus, scaffolds of 8,234–8,303 nt and having up to four short internal gaps and missing short terminal sequences were also assembled from samples 13–02, 13–03, and 13–04. In parallel, besides the determined complete genome sequence of the novel potyvirus, scaffolds of 10,318–10,360 nt and containing up to four short internal gaps were also assembled from the other infected samples (13–03, 13–04, 13–05).

## Genomic organization and phylogenetic relationships of the novel potyvirus

The potyviral genome determined from sample 13–06 is 10,540 nt excluding the poly (A) tail and encodes a polyprotein of 3,210 amino acids (aa) (Fig 2A). The 5' non coding region (NCR) is 159 nt long, whereas the 3' NCR is 751 nt long, which is significantly longer than for most potyviruses [13]. Based on the conserved cleavage sites in the polyprotein sequence [14], the ten typical mature potyviral proteins could be identified with estimated sizes of 422 aa (P1), 456 aa (HC-Pro, helper component proteinase), 359 aa (P3), 52 aa (6K1), 635 aa (CI, cylindrical inclusion protein), 53 aa (6K2), 192 aa (VPg, viral genome-linked protein), 242 aa (NIa, nuclear inclusion a), 513 aa (NIb, nuclear inclusion b), and 286 aa (CP, coat protein). The observed cleavage sites in the polyprotein sequence were consistent with the known sites of potyviruses (Fig 2A). In addition, a PIPO ORF (69 aa) was identified downstream of the conserved slippage motif GAAAAAA (nt position 3283). All expected potyviral conserved motifs were identified in the polyprotein, including in the HC-Pro the KITC (aa position 472 to aa 475) and PTK (730 to 732) and in the CP the conserved DAG that are all necessary for aphid transmission [15].

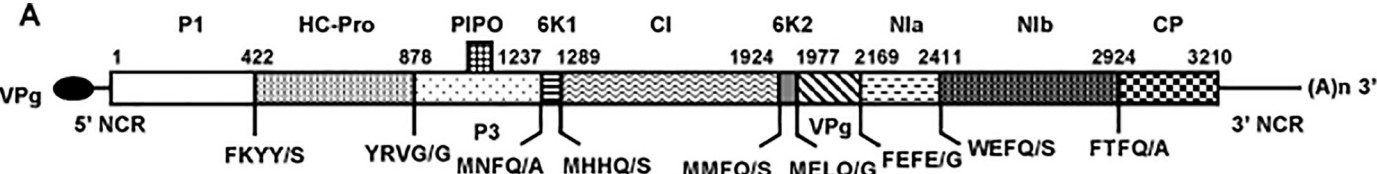

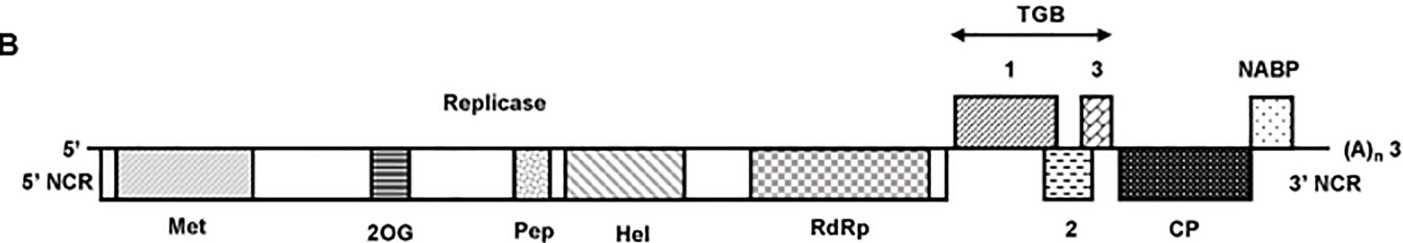

**Fig 2. Shematic representation of the genomic organization of the novel potyvirus (A) and the novel carlavirus (B).** The open reading frames are depicted by large boxes, and the non coding regions (5' and 3' NCR) by horizontal lines. (A)$_n$: PolyA tail. (A) The nine putative cleavage sites of the polyprotein are indicated, as well as the predicted amino acid position for each mature protein in the polyprotein. P1, helper component proteinase (HCPro), P3, 6K1, cylindrical inclusion (CI) protein, 6K2, viral genome-linked protein (VPg), nuclear inclusion a (NIa), nuclear inclusion b (NIb), and coat protein (CP). The position of PIPO (Pretty Interesting Potyviridae ORF) is also indicated. The black ellipse represents the VPg attached to the 5' end of the genome. (B) Conserved motifs for viral methyltransferase (pfam 1660, Met), 2OG-Fe(II) oxygenase (pfam 03171, 2OG), peptidase C23 (pfam 05379, Pep), viral helicase 1 (pfam 01443, Hel), and RNA-dependent RNA polymerase 2 (pfam 00978, RdRp) domains are shown within replicase. TGB 1, 2, 3, Triple gene block proteins 1, 2, and 3. CP, coat protein. NABP, nucleic acid binding protein.

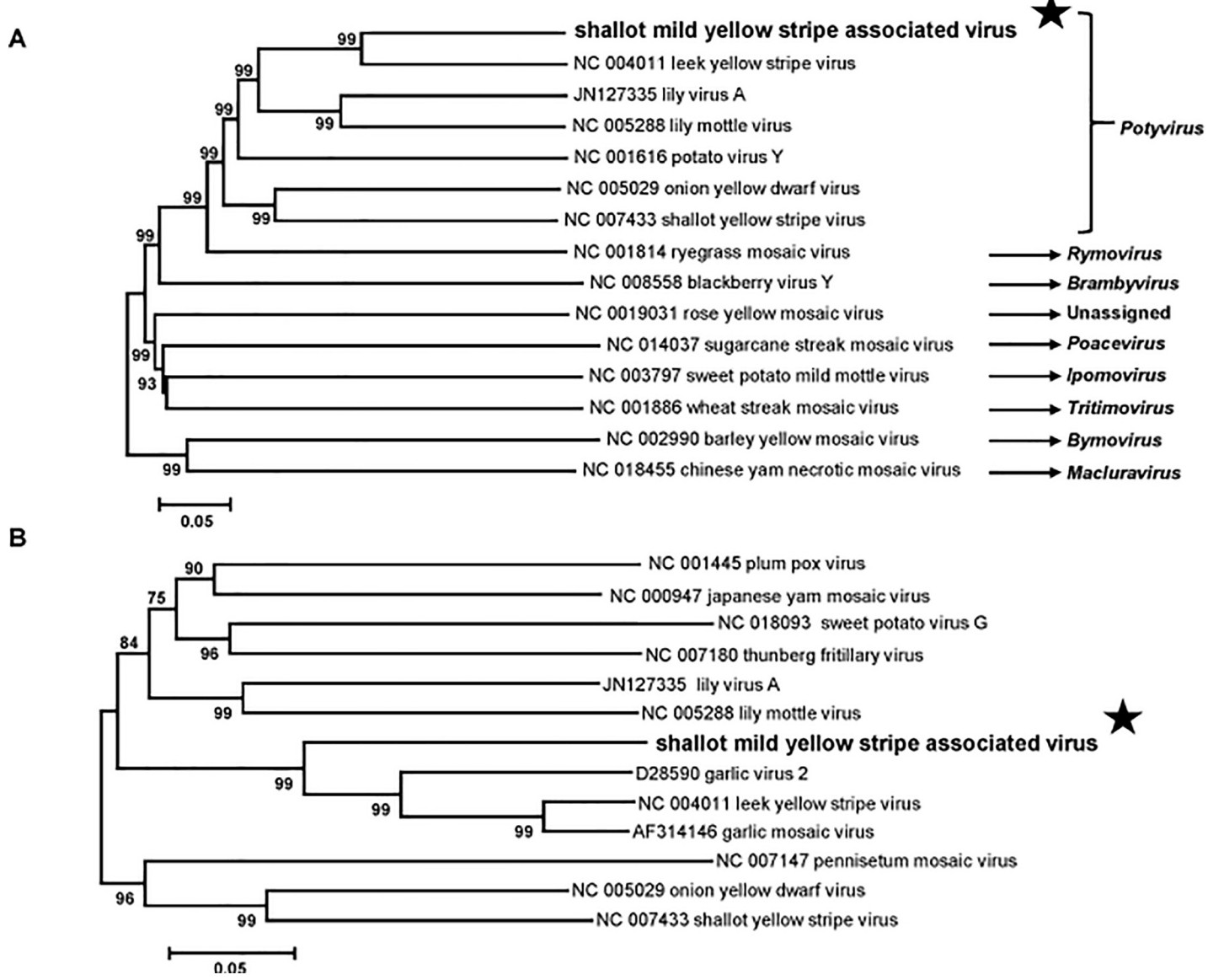

**Fig 3. Unrooted phylogenetic trees based on the codon-aligned nucleotide sequences of the 3' part (from P3 to coat protein) of the polyproteins of representative** *Potyviridae* **family members (A) and on the coat protein sequences of representative members of the genus** *Potyvirus* **(B).** The trees were constructed using the neighbor-joining method and statistical significance of branches was evaluated by bootstrap analysis (1,000 replicates). Only bootstrap values above 70% are shown. The scale bar represents 5% nucleotide divergence (A) or 5% amino acid divergence (B). The genus to which each virus belongs is indicated at the right of the panel A. The novel potyvirus shallot mild yellow stripe associated virus is indicated by a black star.

In order to determine the taxonomic relationships of this virus, a phylogenetic tree was reconstructed using the genomic sequences of representative members of the family *Potyviridae* (from P3 to CP genes, corresponding to the RNA1 of bymoviruses, Fig 3A). Sequence comparisons were then performed with the polyprotein, the coat protein, and the NIa-Pro-NIb genomic region and corresponding proteins of members of the genus *Potyvirus* (S2 Table). The accepted molecular species demarcation criteria for the family *Potyviridae* are less than 76% nt identity or 82% aa identity in the large ORF or its protein product [16]. By all the criteria, the detected potyvirus appears to be a distinct species, with clearly more distant identity levels with its closest fully sequenced relative, LYSV [at the best 68.8% nt identity in the

large ORF (73.6% aa); (S2 Table)]. The name of shallot mild yellow stripe associated virus (SMYSaV) is therefore proposed for this novel potyvirus.

In a phylogenetic analysis of the CP amino acid sequences, SMYSaV forms a small bootstrap-supported cluster with garlic virus 2, leek yellow stripe virus, and garlic mosaic virus (Fig 3B), forming a small group of agents of similar host specificity as observed for other potyviruses. The closest sequence to SMYSaV identified through GenBank Blast searches is a partial, 2,525 nt genome fragment (GenBank accession number L28079) corresponding to the partial protease (NIa-Pro) and RNA-dependent RNA polymerase (NIb) genes of a viral isolate from shallot (unpublished GenBank sequence). Over this region, the two agents show 92.3% nt (94.3% aa) identity (S2 Table), indicating that they belong to the same species. Remarkably the L28079 sequence was described in GenBank database as "shallot potyvirus (probably Onion yellow dwarf virus)" indicating that SMYSaV had been observed previously in shallot in Russia but that its originality and distinctness had not been recognized at the time.

The four symptomatic samples analyzed by HTS in the present study were all found to be infected by SMYSaV (Table 1), allowing the reconstruction of long scaffolds for each isolate. Comparison of these four sequences provides nt identity values ranging from 97.1% and 99.7% (data not shown), giving a first vision of the diversity of this novel virus. Pairwise comparisons showed that the most closely related isolates were from 13–05 and 13–06 plants which have been collected in different fields, and the most distant isolates were from 13–05 and 13–03 plants also collected in different locations.

## Genomic organization and phylogenetic affinities of the novel carlavirus

Widely different amounts of carlaviral reads were detected in four of the six samples analyzed by HTS (one asymptomatic and three symptomatic, Table 1). The genomic sequence was completed for the sample showing the deepest coverage (sample 13–05, representing 22.7% of the total reads). A unique contig, 8,343 nt-long and only missing a short region at the 3' end (as judged by comparison with SLV), was reconstructed. The 5' end was confirmed and the 3' end was determined by RACE experiments. The genome organization is typical of members of the genus *Carlavirus*, with six ORFs encoding from 5' to 3' the viral replicase (REP), the triple gene block proteins (TGB1, 2, 3) involved in viral movement, the coat protein and, finally, a nucleic acid binding protein, whose role is still unclear (Fig 2B). The sizes of the deduced proteins are identical to those of the most closely related carlavirus (SLV), with the exception of the replicase which is slightly larger than in SLV (1,926 aa *vs* 1,924 aa) with 12 indels located in the first part of the deduced protein (data not shown). The conserved motives typical for carlaviral REPs [17] were identified, including a viral methyltransferase domain (pfam 1660, aa 42–352), an AlkB (2OG-FeII-Oxy-2) domain (pfam 03171, aa 681–769), a peptidase C23 (carla endopeptidase) domain (aa 930–1015), a viral helicase 1 domain (pfam 01443, aa 1108–1380) and a RNA-dependent RNA polymerase 2 domain (pfam 1505–1913, aa 1505–1913).

The taxonomical position of the novel carlavirus was confirmed by phylogenetic analyses performed with complete genome sequences of representative members of the families *Alphaflexiviridae* and *Betaflexiviridae* (Fig 4) and with replicase and coat protein sequences from a range of carlaviruses (data not shown). As shown in Fig 4, the carlavirus unambiguously clusters with related members in the family *Betaflexiviridae*. In this and in the other two trees (not shown), it clusters together with SLV with 100% bootstrap support, making SLV its closest relative in the genus. However, the level of identity between SLV and the novel carlavirus in replicase and coat protein genes (and deduced proteins) is clearly below the species demarcation threshold accepted for the family *Betaflexiviridae* (72% nt or 80% aa identities in replicase or CP genes) [18]. Indeed, it shares at the best 76.5% of aa identity in the CP with SLV (69.6% nt

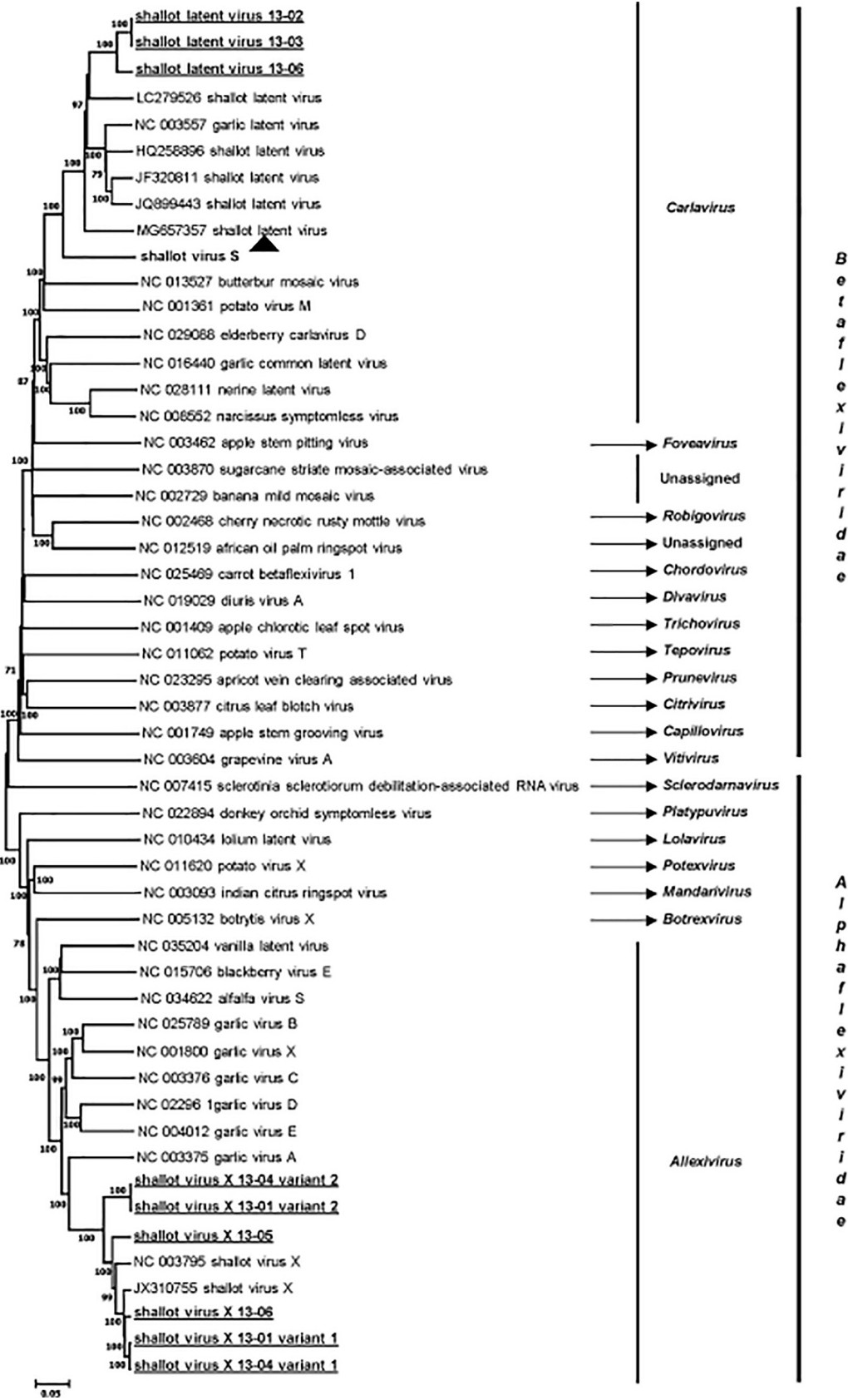

**Fig 4. Neighbor-joining phylogenetic tree reconstructed from the alignment of complete genome sequence of representative members of the families *Alphaflexiviridae* and *Betaflexiviridae*.** Statistical significance of branches was evaluated by bootstrap analysis (1,000 replicates) and only values above 70% are indicated. The scale represents 5% nucleotide divergence. The genus and the family to which each virus belongs are indicated at the right of the figure. The sequences of shallot virus X and shallot latent virus determined in this work are underlined, and the novel carlavirus shallot virus S is indicated by a black triangle.

identity, S3 Table), demonstrating that it represents a novel species in the genus *Carlavirus*, for which the name of shallot virus S (ShVS) is proposed.

## Analysis of the shallot virus X isolates identified by HTS

Six ShVX isolates were identified from one asymptomatic sample and three symptomatic ones. Three full genome sequences and three for which very long contigs, lacking only genome ends, were reconstructed (Table 1). The phylogenetic analysis based on the alignment of the complete genome sequences of *Alphaflexiviridae* members clearly shows that all the sequences reported here belong to the *Shallot virus X* species, forming a cluster supported by a high bootstrap value (Fig 4). The phylogenetic analysis based on the CP sequences of allexivirus members and of the available ShVX isolates retrieved from GenBank confirmed this conclusion (S1 Fig). Moreover, the sequences reported here shared between 79.9% and 97.5% of nt identity (87% to 98.9% aa identity) in the CP gene with reference isolates (data not shown), levels of identity which are within the molecular species demarcation criteria accepted for the family *Alphaflexiviridae* [18]. This conclusion is confirmed by similar analyses performed with polymerase sequences (data not shown). In the CP tree (S1 Fig), four isolates (13–01 variant 1, 13–04 variant 1, 13–05, and 13–06) belong to a cluster comprising six already known ShVX isolates including the only available shallot mite-borne latent virus sequence, which should probably be considered a synonym of ShVX [19]. On the other hand, the two other isolates (13–01 and 13–04 variant 2) form a divergent cluster, away from other known ShVX isolates and from the isolates found in co-infection in the same original plants (S1 Fig). These two isolates are very closely related (99.9% nt identity in the CP gene) and more distant from other isolates (83.5% to 85.5% nt identity, depending on the isolate considered), including the highly divergent Dindugal isolate GQ268322, 80.2% nt identity).

## Analysis of the shallot latent virus isolates identified by HTS

SLV was identified in three samples, one asymptomatic and two symptomatic ones. Complete genome sequences were determined from two of them (SLV 13–02 and 13–06) while for the remaining isolate (SLV 13–03), a very long contig missing only 84 nt and 34 nt at the 5' and 3' ends, respectively was obtained. The isolates analyzed here clearly cluster in the *Shallot latent virus* species (Fig 4) but form a distinct and novel cluster well separated from other known isolates of the virus. For the CP gene, the diversity between them and other SLV isolates ranges between 17.6 and 24.3% in nt (between 5.1 and 10.1% in aa). Although significant, these values are well within the species demarcation criteria for the *Betaflexiviridae* family [17]. The three isolates of SLV analyzed here are very closely related to each other with nt identity levels comprised between 93.7 and 100% in the CP gene (99 to 100% in aa for deduced proteins, data not shown). Similar values are observed in the REP gene (90.9 to 99.8% in nt, 96.1 to 99.8% in aa).

## Host range of both novel viruses and Koch's postulates

Trials to mechanically transmit ShVS to herbaceous dicot plants (*N. benthamiana*, *N. occidentalis*, *C. quinoa* and *C. amaranticolor*) were unsuccessful. Similar negative results were

obtained with SMYSaV: no symptoms were visible on any of the SMYSaV-inoculated plants and no virus could be detected by a specific RT-PCR assay in any of the inoculated dicot hosts.

We then tried to fulfill Koch's postulates, either using each novel virus alone or using a viral complex composed of ShVX, SLV and the two novel viruses. A pool of four plants known to harbor this complex was used to inoculate a total of 21 shallot plants. Most of the inoculated plants (14/21) were found to be co-infected by the four viruses, but no symptoms could be observed in any of the inoculated plants. Concerning the inoculation of the novel viruses alone, ShVS was detected in 100% of the inoculated onion plants and in 29/36 of the inoculated shallots. After five weeks of observation, no symptoms were recorded on inoculated plants, an observation in line with the finding of ShVS in one of the asymptomatic plants analyzed by HTS (Table 1). Similarly, SMYSaV was detected in 75% of the inoculated shallots and in 100% of the inoculated onion plants. However, no symptoms could be observed in the infected plants. As a positive control, leaves from OYDV-infected shallots, but free of SMYSaV, were used to inoculate shallot and onion plants. Five weeks after inoculation, typical yellow mosaic symptoms were observed on both hosts and OYDV was detected in the symptomatic plants by specific RT-PCR (data not shown). Bulbs from all SMYSaV-inoculated shallot plants were replanted and most of the resulting (19/20) plants were found to be infected by SMYSaV, showing that the virus accumulates in the bulbs and can perpetuate the infection over seasons. However, neither the first generation nor second generation plants displayed symptoms under our greenhouse conditions, even after eight months of observation.

## Correlation between virus presence and the symptoms associated with the shallot mild yellow stripe disease

Despite the negative results of the Koch's postulate trials, which do not allow to conclude about a causal role of SMYSaV, the results of the HTS analyses strongly suggest its involvement in the disease, since it is the only virus that was specifically associated with the four symptomatic plants analyzed (Table 1). In order to try to confirm an association between SMYSaV and the SMYS disease symptoms, a correlative analysis involving a large number of plants was performed. Over a period of four years, a total of 351 shallot samples originating from the same region of France were analyzed for the presence of SMYSaV, LYSV and OYDV using specific RT-PCR assays (S1 Table). Twenty-two samples were found to be infected by OYDV or/and LYSV, with a mean of striping score of 2.43 ± 1.03 and a mean score of 1.5 ± 1.46 for the loss of vigor. In the remaining samples, the incidence of SMYSaV was found to be quite high (27.2%) and was highly correlated with the presence of striping symptoms. Indeed, 92.9% of the samples with stripes (score between 1 and 3) were infected by SMYSaV (78/84), whereas 95.9% of the asymptomatic samples were SMYSaV-free (235/245). The mean score of stripe symptoms for the SMYSaV-infected samples (2.39 ± 0.98) was not significantly different from that of OYDV/LYSV-infected samples (2.43 ± 1.03) (Fig 5), indicating that SYMSaV could have the same impact on infected plants in terms of striping severity than the two other potyviruses OYDV and LYSV. In contrast, the effect of SYMSaV infection regarding the loss of vigor is significantly lower (p = 0.0004) than that of OYDV/LYSV infection (Fig 5), strongly suggesting that the symptoms of the SMYSD consisting of yellow stripes on leaves and moderate loss of vigor are associated with SMYSaV.

On a smaller number of analyzed plants (45), the potential contribution of ShVS to the symptomatology was also assessed. The prevalence of this virus was found to be high in the analyzed samples (53.3%) but the infection was not correlated with symptomatology. Indeed, the same proportion of symptomatic or asymptomatic samples from the correlative study were found to be infected by ShVS (48.8% *vs* 50%, respectively).

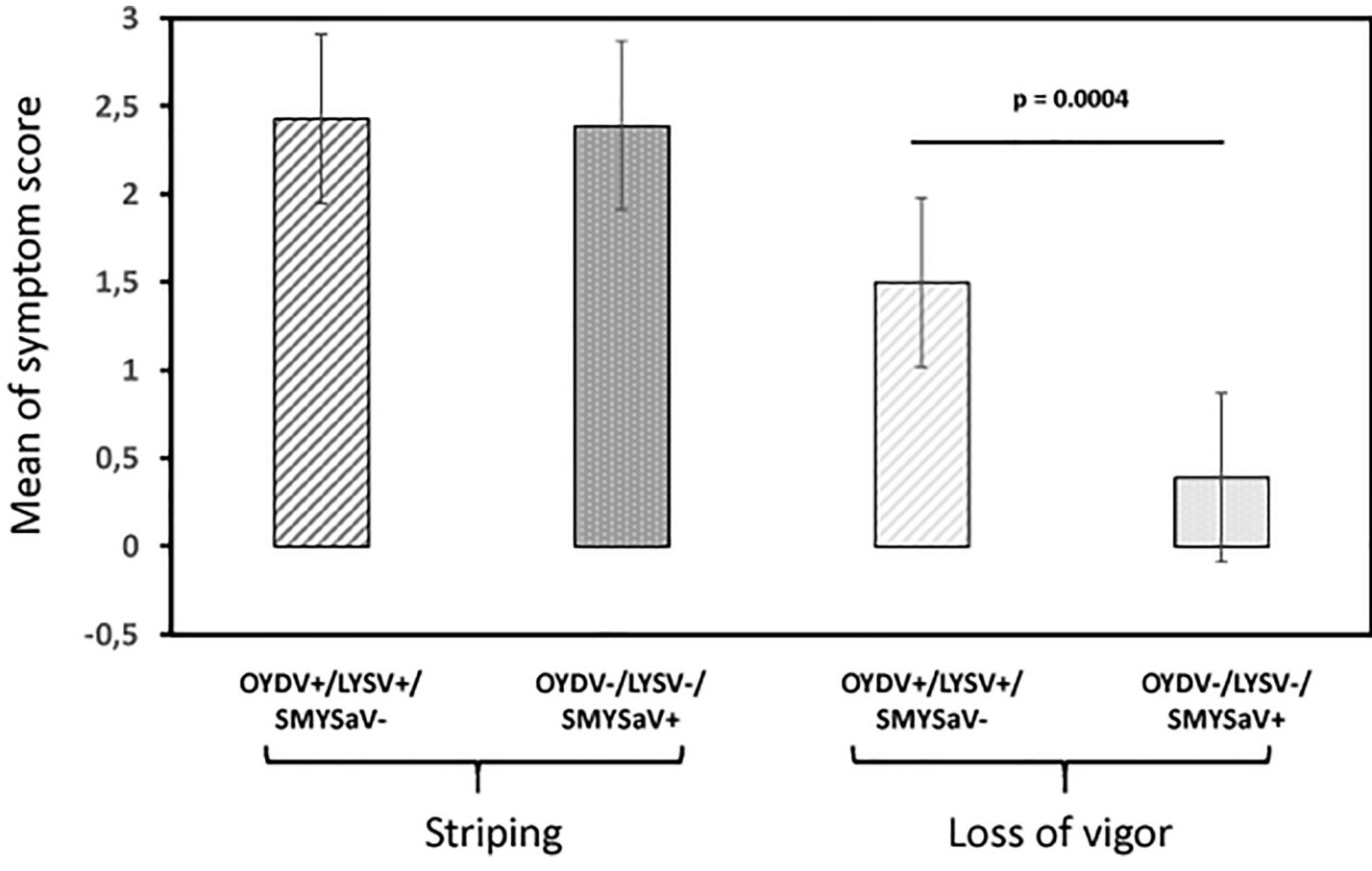

**Fig 5. Comparison of the mean of symptom score (striping and loss of vigor) in two populations of shallot plants.** OYDV+/LYSV+/SMYSaV-: plants infected by onion yellow dwarf virus and/or leek yellow stripe virus and free of shallot mild yellow stripe associated virus. OYDV-/LYSV-/SMYSaV+: plants infected by shallot mild yellow stripe associated virus and free of onion dwarf virus and leek yellow stripe virus. Whiskers indicate the standard error of the mean. The significance (p) was tested by the Mann-Whitney-Wilcoxon non parametric test [20–21].

The diversity of SMYSaV was also analyzed, using the nucleotide sequence of a short fragment of the CP gene targeted by the RT-PCR diagnostic assay. The average pairwise nucleotide divergence was 1.4% between isolates in this region. More interestingly, the diversity could be structured into two distinct clusters, as illustrated by the neighbor-joining tree shown in S2 Fig. Beside the major group (cluster 1) which contains 91% of the isolates, an additional group (cluster 2) could be defined with high bootstrap support (99%). The intra-group average nucleotide divergence is very low (0.7% and 0.5% for clusters 1 and 2, respectively), in comparison with the inter-group average divergence (5.2%). Due to the small number of isolates in the cluster 2, no conclusion could be drawn in terms of correlation between a particular SMYSaV cluster and the severity of the induced symptoms, or the geographical origin of the isolates.

## Discussion

The present study was motivated by reports of a yellow stripe disease on shallot varieties regenerated from OYDV- and LYSV-free bulbs. Subsequent meristem-tip cultures resulted in the clearance of the symptoms, suggesting a viral etiology. The objective of this work was therefore to identify the virus(es) involved in this newly described disease by a combination of HTS-based and classical approaches. Analysis of six shallot samples (two asymptomatic and four

symptomatic) by HTS of purified dsRNAs revealed the presence of two viruses already known to give asymptomatic infections in shallot (SLV and ShVX) and of two novel viruses: a carlavirus named ShVS and a potyvirus named SMYSaV. A partial sequence was already available in GenBank (L28079) for the potyvirus, reported with an uncertain taxonomy as probably belonging to the *Onion yellow dwarf virus* species. The determination of the complete genome sequence and phylogenetic analyses clearly show that this potyvirus is a novel species distinct from OYDV. The genomic organization of ShVS and SMYSaV are similar to those of *Carlavirus* and *Potyvirus* genera members, respectively. Interestingly, the 3' NCR of SMYSaV with a size of 751 nt is significantly longer than reported for potyviruses (around 220 nt, [13]). Other potyviruses belonging to the same phylogenetic cluster (Fig 3B) share this property, with a 3' NCR size of 592 nt for LYSV and 598 nt for garlic virus 2 (no data available for garlic mosaic virus). The biological significance of this observation remains unclear, if any. Nevertheless, the role of the 3' NCR as a determinant of symptom induction has been proposed in a few examples [22], without any hypothesis about the mechanism(s) involved [23].

The novel potyvirus described here is the sole detected virus associated with the symptoms of the SMYSD. SMYSaV infection is strongly associated with striping symptoms, with a severity comparable to those caused by OYDV and/or LYSV infection; on the other hand, the impact of SMYSaV infection in terms of loss of vigor is moderate, as reported for the SMYSD, and quite different from the more severe loss of vigor associated with OYDV and/or LYSV infection (Fig 5). The four symptomatic samples analyzed by HTS were infected with a complex of viruses, which is coherent with the strictly vegetative mode of propagation of shallot. Depending on the sample, various combinations of agents were found, involving SMYSaV and ShVS, ShVX and/or SLV (Table 1). On this basis, trials to fulfill Koch's postulates were pursued involving either SMYSaV alone, ShVS alone or a complex of the four viruses found in symptomatic shallots. However, even over a long period of observation, no symptoms were observed on any of the inoculated shallots, even if most of them were found to be infected by the virus(es) they had been inoculated with. Two hypotheses can be proposed to explain the failure to observe symptoms on the inoculated plants, one is that the greenhouse conditions used would not allow the development of such symptoms. The other is that the shallot variety used in these experiments (a seed-propagated variety, different from the bulb-propagated ones in which the disease is described) may not be conducive to symptoms.

In the HTS-based analysis, the novel carlavirus ShVS was detected in three symptomatic samples as well as in an asymptomatic one, suggesting that as for other shallot infecting carlaviruses, its infection is latent. This hypothesis is confirmed by the finding that the virus was equally distributed between symptomatic and asymptomatic plants in the correlation study. Our results do not allow us to conclude regarding a potential synergistic effect of ShVS with SMYSaV infection, as shown for SLV and GarCLV with potyviruses [2]. Overall, the very tight correlation between SMYSaV infection and the SMYSD symptoms support the notion of an association if not a causal role for SMYSaV, but further experiments are necessary to unambiguously demonstrate it and to explore potential synergistic effects with other co-infecting viruses.

## Supporting information

**S1 Fig. Neighbor-joining tree reconstructed from the alignment of amino acid sequences of the coat protein of allexivirus members and shallot virus X isolates.** Validity of branches was evaluated by bootstrap analysis (1,000 replicates). Only bootstrap values above 70% are shown. The scale bar represents 5% amino acid divergence. The sequences of ShVX determined in this work are underlined. Potato virus X (NC011620, genus *Potexvirus*) was used as

outgroup.
(PDF)

**S2 Fig. Neighbor-joining tree reconstructed from the alignment of nucleotide sequences of a partial fragment (247 nt) of the coat protein gene obtained from a range of shallot mild yellow stripe associated virus isolates.** Statistical significance of the branches was evaluated by bootstrap analysis (1,000 replicates). Only bootstrap values higher than 70% are indicated. The scale bar represents 5% nucleotide divergence. The primer pair used for the RT-PCR (ShMYSV-F1/ShMYSV-R1) is indicated in S1 Table. Relevant nucleotide sequences were deposited in the GenBank database under accession numbers MG910501 to MG910598. Isolates found in co-infection with onion yellow dwarf virus or leek yellow stripe virus are indicated in italics. The scores of leaves striping (S) and loss of vigor (V) are indicated (scale of notation from 0 to 3). The two identified phylogenetic clusters are indicated on the right of the figure. (PDF)

**S1 Table. Oligonucleotides used in the present study for the completion of the seven viral genomes and the detection of onion dwarf virus and leek yellow stripe virus.** (DOCX)

**S2 Table. Percentages of identities in nucleotides (nt) and in amino acids (aa) between shallot mild yellow stripe associated virus and members of the genus *Potyvirus* over the large ORF and in two genomic regions.** (DOCX)

**S3 Table. Percentage of identity in the replicase and coat protein genes and deduced proteins between shallot virus S and closest relative carlaviruses [a].** (DOCX)

## Acknowledgments

The authors would like to thank the GenomEast platform (Institut de Génétique et de Biologie Moléculaire et Cellulaire, CNRS/INSERM/Université de Strasbourg, Illkirch, France) for the Illumina sequencing, and T. Mauduit and C. Chesseron for taking care of experimental plants. The sequences reported in the present manuscript have been deposited in the GenBank database under accession numbers MG571549, MH292861, MH389247 to MH389255, and MG910501 to MG910598.

## Author Contributions

**Conceptualization:** Armelle Marais.

**Investigation:** Chantal Faure, Sébastien Theil.

**Methodology:** Armelle Marais.

**Supervision:** Armelle Marais.

**Validation:** Thierry Candresse.

**Writing – original draft:** Armelle Marais.

**Writing – review & editing:** Thierry Candresse.

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
