## [Decision Letter · Decision Letter 0]

28 Jun 2019

PONE-D-19-14594

Characterization of the virome of shallots affected by the shallot mild yellow stripe disease in France

PLOS ONE

Dear Dr. Marais,

Thank you for submitting your manuscript to PLOS ONE. After careful consideration, we feel that it has merit but does not fully meet PLOS ONE’s publication criteria as it currently stands. Therefore, we invite you to submit a revised version of the manuscript that addresses the points raised during the review process.

As you see the reviewers' report, both reviewers recommended your manuscript for publication in PLOS ONE with minor revision. Both reviewers provided some useful suggestions to improve your manuscript. I suggest you to consider all reviewers comments to include in your revised manuscript. If you do not agree with any of reviewer' comments, please explain so why you do not agree with reviewers' suggestions in your 'response to reviewers' comments file'. 

We would appreciate receiving your revised manuscript by Aug 12 2019 11:59PM. To enhance the reproducibility of your results, we recommend that if applicable you deposit your laboratory protocols in protocols.io, where a protocol can be assigned its own identifier (DOI) such that it can be cited independently in the future. For instructions see: http://journals.plos.org/plosone/s/submission-guidelines#loc-laboratory-protocols

We look forward to receiving your revised manuscript.

Kind regards,

Satyanarayana Tatineni, Ph.D

Academic Editor

PLOS ONE

Journal Requirements:

Reviewers' comments:

Reviewer's Responses to Questions

**Comments to the Author**

1. Is the manuscript technically sound, and do the data support the conclusions?

Reviewer #1: Yes

Reviewer #2: Yes

2. Has the statistical analysis been performed appropriately and rigorously? 

Reviewer #1: N/A

Reviewer #2: N/A

3. Have the authors made all data underlying the findings in their manuscript fully available?

Reviewer #1: No

Reviewer #2: Yes

4. Is the manuscript presented in an intelligible fashion and written in standard English?

Reviewer #1: Yes

Reviewer #2: Yes

5. Review Comments to the Author

Reviewer #1: This publication combines the discovery of two new viral species candidates and the addition of nearly whole genome from known viruses. The study includes high throughput sequencing analysis with downstream efforts to characterize the newly identified viruses through greenhouse assays and field survey. This work is fully acceptable for publication in the journal and some modifications could be added to the document.

The discussion might be too long as it often repeats the observed results. The discussion could therefore be shortened underlining key elements of discussion (for example NCR discussion, the explanation on failure to reproduce symptoms…).

The consensus sequences have a reference number in Genebank but the raw data have not been made publicly available on dedicated repository (unless I missed it). It is preferable to have them available for the scientific community.

Line 59 : for the known Allium carla- and potyviruses : are they transmitted by the same species of aphids and which ones ?

Line 59: is it the viruses or the aphids that are frequently found on cultivated Allium species ?

Line 79:which proportion/number of plants were negative for OYDV and LYSV: it was all the plants or a majority ? This orientated the selection of samples for sequencing ?

Line 93: were the samples from the same location/field; neighboring plants ?

Line 150: the host range test corresponds to the ability to infect indicator plants using mechanical inoculation and not closely related plant species by aphid transmission mode. I would change the title accordingly. Host range test would rather be transmission by aphids on different Allium species

Line 165 : Koch’s postulate. Could you describe with more details the inoculation protocol for the 21 virus-free shallots. Was it mechanical or by aphids ?

Line 175: the positive control is on shallot or onion or both ?

Line 177-179: the planting was done in field or in insect-proof greenhouse ?

Line 305: a first view on the diversity of the virus is indeed obtained but more details on the geographical localisation of isolates could be informative

Line 353: the genomes have been sequenced from symptomatic and/or asymptomatic samples ? There is the information in Table 1 but this could be underlined here.

Line 377: same question as before: symptomatic or asymptomatic samples ?

For the analysis of shallot virus X, shallot latent virus and SMYSaV (Fig S2 for the latter), is there any geographical explanation in the clustering of isolates for whole genome or for CP only ? The tw clusters of SMYSaV are they geographically different ?

Reviewer #2: The MS titled “Characterization of the virome of shallots affected by the shallot mild yellow stripe disease in France” describes the effort to determine the disease agent of a new disease of shallots. The authors identify the viruses in diseased sampled using a HTS approach, then verify the genomes with completion of the UTRs using RACE. They also screen samples collected over a time period (symptomatic and asymptomatic) for these and other known shallot viruses. They also attempt to fulfil the Koch’s postulate for the some of the new viruses identified in this study.

In general, I do not see any issues with the MS. It is clear and well written. There are section that one can reduce by making some summarized in tables (% identities etc). I only have minor edits for the authors and these I have made on the attached pdf.

6. PLOS authors have the option to publish the peer review history of their article (what does this mean?). If published, this will include your full peer review and any attached files.

Reviewer #1: Yes: Sebastien Massart

Reviewer #2: No

---

## [Author Response · Author response to Decision Letter 0]

3 Jul 2019

Response to Reviewers

In order to take into account the suggestions made by the reviewers, the manuscript has been modified as detailed below in red following each referee comment.

Reviewer #1:

This publication combines the discovery of two new viral species candidates and the addition of nearly whole genome from known viruses. The study includes high throughput sequencing analysis with downstream efforts to characterize the newly identified viruses through greenhouse assays and field survey. This work is fully acceptable for publication in the journal and some modifications could be added to the document.

The discussion might be too long as it often repeats the observed results. The discussion could therefore be shortened underlining key elements of discussion (for example NCR discussion, the explanation on failure to reproduce symptoms…).

We agree, and we have shortened the discussion by deleting the part concerning the diversity of ShVX and SLV, and some sentences on SMYSaV.

The consensus sequences have a reference number in Genebank but the raw data have not been made publicly available on dedicated repository (unless I missed it). It is preferable to have them available for the scientific community.

The raw data are now available on portail data INRA at https://doi.org/10.15454/6XCMAE. We have added this information in the Material and Methods.

Line 59 : for the known Allium carla- and potyviruses : are they transmitted by the same species of aphids and which ones?

Some aphid species are known to transmit Allium carla- and potyviruses as well, such as Myzus persicae which is able to transmit the potyviruses OYDV and LYSV, and the carlavirus SLV, or M. ascalonicus which transmits OYDV and SLV. Nevertheless, there are some aphid species not able to transmit both virus genera, as well.

Line 59: is it the viruses or the aphids that are frequently found on cultivated Allium species ?

The sentence has been clarified in the text. It is the viruses that are frequently found on cultivated Allium species. In most situations, Allium crops are not found infested with aphids, which visit the crops only occasionally.

Line 79:which proportion/number of plants were negative for OYDV and LYSV: it was all the plants or a majority ? This orientated the selection of samples for sequencing ?

Around 95% of plants showing the symptoms of shallot mild yellow stripe disease were found to be negative for OYDV and LYSV. The samples analyzed by HTS were first screened for the presence of OYDV and LYSV and four negative ones were selected for HTS analysis.

Line 93: were the samples from the same location/field; neighboring plants ?

The two symptomatic samples 13-04 and 13-05 have been collected in the same field, while the four remaining samples (one symptomatic and the two asymptomatic ones) were from distinct location. We have added this information in the text.

Line 150: the host range test corresponds to the ability to infect indicator plants using mechanical inoculation and not closely related plant species by aphid transmission mode. I would change the title accordingly. Host range test would rather be transmission by aphids on different Allium species

We agree. We have changed the title of this paragraph to “Mechanical inoculation of both novel viruses to indicator plants”.

Line 165 : Koch’s postulate. Could you describe with more details the inoculation protocol for the 21 virus-free shallots. Was it mechanical or by aphids ?

We have clarified this point in the text: “A total of 21 virus-free shallots grown from seeds were mechanically inoculated with a mix of four plants shown to be co-infected by ShVX, SLV and the novel carlavirus and potyvirus, as described above.”

Line 175: the positive control is on shallot or onion or both ?

The positive control was used to inoculate both onion and shallot plants. We have added this information in the text.

Line 177-179: the planting was done in field or in insect-proof greenhouse ?

Bulbs from all inoculated shallot plants were replanted in insect-proof greenhouse. We have added this information in the text.

Line 305: a first view on the diversity of the virus is indeed obtained but more details on the geographical localisation of isolates could be informative

Pairwise comparisons showed that the most closely related isolates were from 13-05 and 13-06 plants which have been collected in different fields, and the most distant isolates were from 13-05 and 13-03 plants also collected in different locations. We have added this sentence in the manuscript.

Line 353: the genomes have been sequenced from symptomatic and/or asymptomatic samples ? There is the information in Table 1 but this could be underlined here.

The genomes have been sequenced from one asymptomatic sample and two symptomatic ones. This point has been underlined in the text.

Line 377: same question as before: symptomatic or asymptomatic samples ?

We identified SLV in one asymptomatic sample and in two symptomatic ones. We have added this information in the text.

For the analysis of shallot virus X, shallot latent virus and SMYSaV (Fig S2 for the latter), is there any geographical explanation in the clustering of isolates for whole genome or for CP only ? The tw clusters of SMYSaV are they geographically different ?

All the SLV isolates belong to the same cluster, so that this question is not relevant. Concerning ShVX, the clustering of isolates seems to be not correlated with the geographical origin, since the plants 13-04 and 13-01 in which the variant 2 of ShVX was found were not from the same location.

Likewise, the two clusters of SMYSaV can’t be differentiated according to the geographical origin of the isolates ([Supplementary-material pone.0219024.s002]). Indeed, Cluster 1 is composed of isolates from 10 different locations, while Cluster 2 puts together eight isolates originating from four different locations that are shared by cluster 1’s isolates.

Reviewer #2:

The MS titled “Characterization of the virome of shallots affected by the shallot mild yellow stripe disease in France” describes the effort to determine the disease agent of a new disease of shallots. The authors identify the viruses in diseased sampled using a HTS approach, then verify the genomes with completion of the UTRs using RACE. They also screen samples collected over a time period (symptomatic and asymptomatic) for these and other known shallot viruses. They also attempt to fulfil the Koch’s postulate for the some of the new viruses identified in this study.

In general, I do not see any issues with the MS. It is clear and well written. There are section that one can reduce by making some summarized in tables (% identities etc). I only have minor edits for the authors and these I have made on the attached pdf.

All the suggestions have been introduced into the text.

---

## [Editor Report · Decision Letter 1]

10 Jul 2019

Characterization of the virome of shallots affected by the shallot mild yellow stripe disease in France

PONE-D-19-14594R1

Dear Dr. Marais,

We are pleased to inform you that your manuscript has been judged scientifically suitable for publication and will be formally accepted for publication once it complies with all outstanding technical requirements.

With kind regards,

Satyanarayana Tatineni, Ph.D

Academic Editor

PLOS ONE
---

## [Editor Report · Acceptance letter]

17 Jul 2019

PONE-D-19-14594R1

Characterization of the virome of shallots affected by the shallot mild yellow stripe disease in France

Dear Dr. Marais:

I am pleased to inform you that your manuscript has been deemed suitable for publication in PLOS ONE. Congratulations! Your manuscript is now with our production department.

With kind regards,

on behalf of

Dr. Satyanarayana Tatineni

Academic Editor

PLOS ONE